# Clonal Dispersal of *Cryptococcus gattii* VGII in an Endemic Region of Cryptococcosis in Colombia

**DOI:** 10.3390/jof5020032

**Published:** 2019-04-15

**Authors:** Carolina Firacative, Germán Torres, Wieland Meyer, Patricia Escandón

**Affiliations:** 1Studies in Translational Microbiology and Emerging Diseases (MICROS) Research Group, School of Medicine and Health Sciences, Universidad del Rosario, Bogota 111221, Colombia; cfiracative@gmail.com; 2Molecular Mycology Research Laboratory, Centre for Infectious Diseases and Microbiology, Westmead Clinical School, Faculty of Medicine and Health, Marie Bashir Institute for Infectious Diseases and Biosecurity, The University of Sydney, Westmead Hospital (Research and Education Network), Westmead Institute for Medical Research, Westmead, NSW 2145, Australia; wieland.meyer@sydney.edu.au; 3Microbiology Group, Instituto Nacional de Salud, Bogota 11321, Colombia; gertor@gmail.com

**Keywords:** *Cryptococcus gattii*, genotype, MLST, virulence factors, phenotype

## Abstract

This study characterized the genotype and phenotype of *Cryptococcus gattii* VGII isolates from Cucuta, an endemic region of cryptococcal disease in Colombia, and compared these traits with those from representative isolates from the Vancouver Island outbreak (VGIIa and VGIIb). Genetic diversity was assessed by multilocus sequence typing (MLST) analysis. Phenotypic characteristics, including growth capacity under different temperature and humidity conditions, macroscopic and microscopic morphology, phenotypic switching, mating type, and activity of extracellular enzymes were studied. Virulence was studied in vivo in a mouse model. MLST analysis showed that the isolates from Cucuta were highly clonal, with ST25 being the most common genotype. Phenotypically, isolates from Cucuta showed large cell and capsular sizes, and shared phenotypic traits and enzymatic activities among them. The mating type a prevailed among the isolates, which were fertile and of considerable virulence in the animal model. This study highlights the need for a continuous surveillance of *C. gattii* in Colombia, especially in endemic areas like Cucuta, where the highest number of cryptococcosis cases due to this species is reported. This will allow the early detection of potentially highly virulent strains that spread clonally, and can help prevent the occurrence of outbreaks in Colombia and elsewhere.

## 1. Introduction

Cryptococcosis, a systemic mycosis that affects both human and animals, is caused by the encapsulated yeasts of the *Cryptococcus neoformans* and *C. gattii* species complexes. The first complex has long been separated into two varieties, *C. neoformans* var. *grubii* and *C. neoformans* var. *neoformans*, corresponding to the serotypes A and D, respectively, while *C. gattii* has not been divided into varieties but comprises the serotypes B and C [1]. Among both species, hybrid isolates have been recognized, including serotype AD hybrids, which are the most commonly recovered, and the rarely isolated AB and BD hybrids [2]. Using molecular techniques, such as PCR-fingerprinting [3], restriction fragment length polymorphism (RFLP) [4], and multilocus sequence typing (MLST) [5], isolates of *C. neoformans* and *C. gattii* have in turn been grouped into eight major molecular types, VNI (including VNB) to VNIV for *C. neoformans* and VGI to VGIV for *C. gattii*. Although these molecular types have recently been suggested to be independent species [6], several researchers working on *Cryptococcus* and cryptococcosis agree that this taxonomic proposal is still premature and without consensus in the community [7]. As such, *C. neoformans* and *C. gattii* are herein referred to with their *sensu stricto* names.

Besides the genetic differences between these two species complexes and eight major molecular types, their geographical distribution differs considerably, and it is associated with the number of cases of cryptococcosis reported for each species. While *C. neoformans* has a worldwide distribution and predominates among the isolates (mostly the molecular type VNI), *C. gattii* has a more restricted distribution, which was previously thought to be unique to tropical and subtropical regions, and therefore causes fewer cases [1,8]. However, since the cryptococcosis outbreak reported on Vancouver Island, Canada, in 1999, several *C. gattii* cases have been described from a number of new geographical regions around the world [8,9]. The Vancouver Island outbreak was characterized by a high incidence of cryptococcosis cases in humans, and domestic and wild animals. Just in 2003, as many as 37 cases of cryptococcosis per million inhabitants were reported, which is the highest incidence reported in the world so far [10]. After its occurrence in British Columbia, the dispersion of *C. gattii* into the Pacific Northwest (PNW) of the USA was reported, and since then, this pathogen has been recovered not only in clinical, but also in veterinary and environmental sources in multiple regions in the USA, some of them endemic [9,11]. Genotyping of the strains responsible for the epidemic on Vancouver Island and in the PNW identified all isolates as *C. gattii* VGII, with three subtypes among them, namely, VGIIa, VGIIb, and VGIIc. Among these, VGIIa and VGIIc, the latter reported until now only from Oregon, USA, were characterized for being the most virulent when studied in animal infection models [12,13].

In addition to the abovementioned regions, Australia and Papua New Guinea are also considered endemic for *C. gattii*, although in the latter country its ecological niche remains unknown [14,15]. In South America, countries such as Brazil, Colombia, and Mexico have a considerably high prevalence of *C. gattii* infections and have a significant environmental recovery rate for this species [16]. In Europe, however, only sporadic reports of infection caused by *C. gattii* and an infrequent isolation from the environment have been documented [17,18,19].

In Colombia, the surveillance group for cryptococcosis reported that in the last 20 years, 96.5% of cases were caused by *C. neoformans* var. *grubii* (serotype A), as occurring worldwide [1], 0.4% by *C. neoformans* var. *neoformans* (serotype D), 2.9% by *C. gattii* serotype B and 0.3% by *C. gattii* serotype C [20]. However, in Colombia, the molecular type VGII prevails among *C. gattii* serotype B strains [4,21], although this population is not as genetically diverse as other isolates from South America, mostly from Brazil, which represents the major source of the VGII diversity [22,23]. In a recent MLST study, which included 25 clinical *C. gattii* VGII isolates recovered from 1997 to 2011 from nine departments or administrative subdivisions in Colombia, nine sequence types (STs) were identified, although most isolates (68%), from six departments, belonged to a single sequence type, ST25 [22]. Preliminary studies have confirmed as well that VGII isolates have caused cryptococcosis cases in Colombia since 1990 [4,21]. In addition, in the city of Cucuta, which has the highest incidence rate of cryptococcosis in Colombia (0.56 cases per 100,000 people) [20], the majority of the cases in immunocompetent patients are caused by *C. gattii* serotype B (77%) [24], although only one strain of this serotype has been isolated from the environment in the city [25], indicating a low recovery rate from nature.

Therefore, it is of great importance to characterize, both at the genotypic and phenotypic level, the *C. gattii* isolates responsible for several cryptococcosis cases reported in Cucuta, an endemic region of this mycosis in Colombia. This study also aimed to determine whether these isolates from Cucuta share any traits with the strains responsible for the Vancouver Island cryptococcosis outbreak. Isolates from Cucuta were compared with two well-characterized outbreak strains that represent each of the two major genotypes of this outbreak, VGIIa and VGIIb. This might help explaining the establishment of cryptococcal disease in the region. The outcome of these studies provides important information regarding the epidemiology of this primary pathogen in Colombia, principally in Cucuta, considering the importance of clonal reproduction as a strategy for successful evolution and dispersion, and also provides further knowledge leading to a better understanding of aspects of the pathogenesis of cryptococcosis.

## 2. Materials and Methods

### 2.1. Fungal Isolates

Thirteen clinical *C. gattii* VGII serotype B isolates from Cucuta, recovered over a period of fifteen years (1993–2008) and maintained in the collection of the Instituto Nacional de Salud in Bogota, Colombia, were studied (Table 1). Two *C. gattii* serotype B isolates ENV152 (WM 02.221, representative of VGIIa) and RB28 (WM 02.301, representative of VGIIb) from the Vancouver Island outbreak [10], obtained from the Molecular Mycology Research Laboratory, University of Sydney at Westmead Institute for Medical Research, Australia, were also included to compare phenotypic and genotypic characteristics with strains from a well-reported outbreak, to identify commonalities and/or differences between the two scenarios. From the isolates from Cucuta, six were previously characterized by MLST [22], and from these, one (H0058-I-1278 = WM 05.275) by whole genome sequencing (WGS) [23]. This is the first time that all isolates were characterized by both phenotypic and genotypic methods and compared to traits from those subtypes responsible for the Vancouver Island outbreak.

### 2.2. Reference Strains

*Cryptococcus gattii* strains WM 179 (serotype B, VGI), WM 178 (serotype B, VGII), WM 175 (serotype B, VGIII), and WM 779 (serotype C, VGIV) [4,26] were used as reference strains and also served as internal controls for the reproducibility of all typing techniques. To determine the mating type and mating potential of the studied isolates, the reference strains JEC20 (serotype D, mating type (MAT) a), JEC21 (serotype D, MATα) [27], and the *crg1Δ* mutant derivatives JF109 (MATa) and JF101 (MATα), which have shown a greater expression of pheromone due to the alteration in the MAT locus in the gene *CRG1α* [28], were used.

### 2.3. Molecular Type and Mating Type Determination

High molecular weight DNA was extracted as previously described [29] and the DNA concentration was determined by a spectrophotometer at 260/280 nm. To identify the molecular type of the isolates, the orotidine monophosphate pyrophosphorylase (*URA5*) gene was amplified with the primers: *URA5* (5′-ATGTCCTCCCAAGCCCTCGACTCCG-3′) and SJ01 (5′-TTAAGACCTCTGA ACACCGTACTC-3′) followed by enzymatic restriction with *Sau*96I and *Hha*I, as formerly described [4]. The *URA5*-RFLP patterns were assigned by comparing them to the reference strains of the major molecular types VGI–VGIV [4,21]. Mating type of the isolates were determined as previously reported [30], using specific primers for the amplification of the MATa and MATα locus. Reference strains JEC20 (MATa) and JEC21 (MATα) were used.

### 2.4. Multilocus Sequence Typing (MLST)

The International Society for Human & Animal Mycology (ISHAM) MLST consensus scheme for the *C. neoformans* and *C. gattii* species complexes, including the following seven unlinked genetic loci: *CAP59*, *GPD1*, IGS1, *LAC1*, *PLB1*, *SOD1,* and *URA5*, was used for genotypic analysis. These loci were amplified using the primers and amplification conditions listed at mlst.mycologylab.org [5]. Amplification of the genetic loci was carried out in the Molecular Mycology Research Laboratory, University of Sydney at Westmead Hospital, Westmead, Australia, and the sequences were obtained commercially (Macrogen Inc., Seoul, Korea). Sequences were manually edited with Sequencher 5.2 (Gene Codes Corporation, MI, USA). The MLST gene sequence data for all samples presented herein were deposited and are publicly available at mlst.mycologylab.org. The individual locus sequences and the concatenated sequences were used to generate dendrograms with the program Mega 7.0.26 (Center for Evolutionary Medicine and Informatics, Tempe, AZ, USA) [31], based on maximum likelihood analysis. From previous global MLST studies, data from additional clinical, veterinary, and environmental strains from different parts of the world were selected and included exclusively for the genetic analysis [9,22,23,32,33,34] (Appendix A).

### 2.5. Phenotypic Characterization

The ability of the isolates to grow under different temperatures (0 °C, 15 °C, 25 °C, 37 °C, and 40 °C) was assayed following the protocols previously described [35,36]. To achieve different conditions of relative humidity, a humidity chamber was used, and the ability of isolates to grow under different humidity conditions (20%, 40%, 60%, 80%, and 100%) was assessed as previously described [37,38]. 

Following standardized protocols, the total diameter and the yeast cell diameter of 20 yeast cells per isolate stained with India ink were measured microscopically [36]. Capsule size was estimated as half the difference between the diameter of the total cell and the yeast cell.

The texture (mucoid or smooth and regular or irregular borders) and the diameter of colonies of the isolates grown on Sabouraud dextrose agar (SDA) at 37 °C were evaluated [36]. The capacity of each isolate to change its colony morphology, or phenotypic switching, was also tested on SDA using a previously described protocol [39]. Depending on the growth of the colonies, two to three hundred colonies were observed.

To assess the ability of the isolates to mate, each isolate was plated on V8 media with the opposite mating type. Both JEC20 (MATa) and JEC21 (MATα), as well as JF109 (MATa) and JF101 (MATα), were used as reference strains according to previous studies [28]. The mating type capacity of the isolates was observed using calcofluor white staining and fluorescence microscopy to confirm the presence of sexual structures such as hyphae, basidia, and basidiospores. 

### 2.6. Enzymatic Activities

The activity of the phenoloxydase was detected in L-3,4 dihidroxifenilalanina (L-DOPA) medium using a previously described procedure [35], in which the enzymatic activity is classified according to the pigmentation of the colonies in the following values: 0, negative; 0.5–1, low; 2, medium, and 3-4, high. The phenoloxydase activity was also measured by spectrophotometer [40] at 475 nm in L-DOPA media, adjusting the cell suspension to 1.5 × 10^8^ cells/mL. 

The activity of proteases and phospholipases was assayed in culture media with a yeast carbon base and SDA supplemented with egg yolk, respectively, as previously described [41,42]. Each assay was done in triplicate. The result of each assay was determined through the Pz index (colony diameter divided by the colony diameter plus the hydrolysis zone) [41,42]. The following values of the enzymatic activity were used according to the Pz index: Pz = 1, negative activity; Pz from 0.7 to 0.99, low enzymatic activity; Pz from 0.5 to 0.69, media enzymatic activity; Pz < 0.5, high enzymatic activity [43].

Urease activity was measured in urea broth [43]. Incubation was performed at 37 °C for 144 hours, and measurements of enzyme activity were made by spectrophotometer at 550 nm, after 0, 24, 72, 96, 120, and 144 hours in 96-microwell plates using an ELISA reader (Beckman). The procedures were performed in triplicate.

### 2.7. Virulence Study

The virulence of the four isolates from Cucuta and both Canadian strains was studied in vivo. The Colombian isolates were selected after studying the virulence factors in vitro. Isolates that showed at some extended differences in the expression of the virulence factors were selected. For the animal experiment, 5-week-old female BALB/c mice (weighing between 16 to 20 g) bred at the Animal House of the Instituto Nacional de Salud, Bogota, Colombia, were used. Intravenous (tail vein) inoculation of the mice was done with 100 µL of a yeast suspension adjusted to a concentration of 5 × 10^6^ cells/mL [35]. Per cryptococcal strain, five animals were inoculated and monitored daily for 70 days. When signs of infection, including ruffled fur, inactivity, weight loss, difficulty in breathing, and neurological signs, such as ataxia, were observed, at any point in the experiment, mice were sacrificed by euthanasia with CO_2_ (5%) [35]. During the time of the experiment, mice were kept in standard cages in groups of five with access to water and food ad libitum. As a control of infection, mice inoculated with sterile saline solution at 0.85% were used. The *C. gattii* isolate WM 198, known to be highly virulent [44], was used as a control. Virulence of the isolates was assessed by animal survival and dissemination of the yeast to the following organs: lungs, spleen, and brain. For the histopathological study, necropsy was performed on each mouse, extracting brain, lungs, and spleen, which were placed in amber bottles with 10 mL of 10% formaldehyde and sent to the histopathology laboratory at the National Institute of Health in Colombia. The animal study was performed with the approval of the Ethics Committee of the Instituto Nacional de Salud, Colombia (CTIN17/2006). 

### 2.8. Statistical Analysis

Evaluation of error assumptions: normality was tested using Kolmogorov–Smirnov and Shapiro–Wilk tests, and homogeneity of variance was tested using the Levene test. In those assays where the data met the assumptions of error, a two-tailed *t*-test to evaluate differences between groups of isolates from Colombia and Vancouver was performed. To evaluate differences among isolates, one-way ANOVA was performed, followed by multiple comparisons using the Scheffe and Tukey tests. The assays in which data did not meet the assumptions of error were analyzed by analysis of variance with non-parametric tests using the Kruskal–Wallis and Mann–Whitney tests to compare pairs of isolates tested. For the selection of isolates evaluated in the animal model, principal component analysis was performed to determine which virulence factors had a greater statistical weight. All statistical analyses were performed in SPSS 16.0 (SPSS Inc.). For the in vivo virulence study, median survival times were obtained and estimation of differences in survival was analyzed by the log-rank (Mantel–Cox) test. Statistical analyses and plots were generated using GraphPad Prism version 6.0b (La Jolla, CA, USA), with *p*-values < 0.05 considered statistically significant. 

## 3. Results

### 3.1. *C. gattii* VGII Isolates from Cucuta Were Highly Clonal as Established by MLST

By URA5-RFLP analysis, the molecular type of all 13 isolates from Cucuta was confirmed as VGII, as well as the molecular type of the two representative isolates from the Vancouver Island outbreak, VGIIa and VGIIb, respectively. Among all Colombian isolates, only three sequence types (ST25, ST47, and ST258) were identified using MLST analysis, with ST25 being the most common one (11 out of 13) (Table 2). Nevertheless, from a total of 4164 bp that corresponded to the concatenated sequences of the seven genetic loci, ST47 and ST258 have only 15 and 18 SNPs, respectively, compared to ST25. Compared to the Vancouver Island outbreak strains, both ST25 and ST47 isolates shared the sequences of three genes (*CAP59*, *GPD1*, and *LAC1*) with the VGIIb isolate (ST7), and the isolate from Colombia with ST258 (H0058-I-357) shared the sequences of two other genes (*PLB1* and *URA5*) with the VGIIb isolate (ST7). In addition, the isolate with ST47 (H0058-I-255) shared two other genes (*PLB1* and *URA5*) with the VGIIa isolate (ST20) (Table 2). Overall, the Colombian isolates were more closely related to the VGIIb than to the VGIIa isolates from Vancouver Island, as the distance (dissimilarities) between most isolates from Colombia and the VGIIb isolate is shorter (Figure 1).

Twelve of the 13 *C. gattii* isolates from Cucuta were mating type a (92.3%), and only one isolate (H0058-I-255) was mating type α (7.7%). Both isolates from Vancouver Island were confirmed to be mating type α (Table 2).

### 3.2. Isolates Share Several Phenotypic Traits

The growth of the isolates from Cucuta at different temperatures and moisture conditions was similar among them and did not differ to the growth of the isolates from Vancouver Island. No statistical difference was observed among the studied isolates in the growth rate at 15 °C, 25 °C, and 37 °C (*p* = 0.89), and at 0 °C and 40 °C, no evidence of growth was observed in any of the isolates. For all isolates as well, the growth rate was directly proportional to the percentage of relative humidity, without any statistical difference in the colony diameter among the Colombian isolates (*p* = 0.92) and compared with the isolates from Vancouver Island. On average, the diameter of the colonies of the isolates from Cucuta was 1.35 mm, 1.73 mm, 2.73 mm, 3.35 mm, and 4.48 mm when growing at 20%, 40%, 60%, 80%, and 100% relative humidity, respectively. 

Generally, the Colombian isolates showed a bigger cell and capsular diameter, compared with the two isolates from Vancouver Island (*p* < 0.05) (Figure 2), except for the isolate H0058-I-3030 that presented similar cell and capsular size to the VGIIa isolate (*p* > 0.11). Overall, Colombian strains had an average cell size of 13.03 ± 3.31 µm and an average capsular diameter of 5.93 ± 2.15 µm. The VGIIa isolate from Vancouver Island presented an average cell and capsular size of 6.5 ± 1.26 µm and 2.43 ± 0.28 µm, respectively, while the VGIIb isolate presented an average cell and capsular size of 4.19 ± 1.09 µm and 1.69 ± 0.73 µm, respectively.

At 37 °C, all studied isolates showed regular borders in their colonies with no statistically significant differences in the diameter of their colonies (*p* = 0.41). Isolates from Cucuta presented an average colony diameter of 8.74 mm, and the VGIIa and VGIIb isolates from Vancouver Island an average of 8.63 and 8.3 mm, respectively (Table 3). From the Colombian isolates, 11 (84.6%) showed phenotypic switching capacity as well as both isolates from Vancouver Island (Table 3).

### 3.3. Most Isolates Were Fertile and Mated with Opposite Mating Types

In vitro mating with the JEC21 MATα reference strain showed that eight out of the 12 Colombian *C. gattii* isolates that were classified as MATa by PCR had the ability to express their sexual phase. However, when crossed with the control JF101 MATα strain, mating capability was observed only with three of the Colombian isolates: H0058-I-239, H0058-I-357, and H0058-I-3030. The isolate H0058-I-255, which was determined as MATα by PCR, showed no ability to mate with the control JEC20 MATa strain, although, this strain mated with the control JF109 MATa strain. The Colombian isolates H0058-I-223 and H0058-I-2877 did not express their sexual phase in vitro. Both isolates from Vancouver Island were able to mate with the JF109 MATa strain, showing a sexual phase.

When tested with the strains JEC20 and JEC21, the sexual phase became evident macroscopically on the surface of the Petri dishes, which was evidenced by the presence of hyphae, basidia, clamp connections, and basidiospores under fluorescent microscopy. In contrast, when the isolates were tested with the strains JF109 (MATa) and JF101 (MATα), the presence of these structures could only be seen by observing the cultures on the Petri dishes directly under the microscope at 10×, because most of the mycelium was immersed in the agar. Both isolates from Vancouver Island (MATα) were crossed with the 12 Colombian isolates MATa, and the Colombian isolate H0058-I-255 (MATα) against the 12 MATa Colombian isolates, but no mating was observed between the crosses described above. By fluorescence microscopy, sexual structures of *C. gattii* VGII isolates from Cucuta that mated with opposite mating type strains were evidenced, including basidia, formation, and detachment of basidiospores from basidium (Figure 3).

### 3.4. Enzymatic Activities Slightly Differ Among the Isolates

Qualitatively, it was determined that seven (53.8%) of the isolates from Cucuta have a high phenoloxydase activity, 4 (30.8%) had a medium activity, and 2 (15.4%) had a low activity. As for the Vancouver Island isolates, it was determined that both VGIIa and VGIIb strains have a high activity. Quantitatively, however, it was found that there was a significant variability in the melanin production of the isolates from Cucuta (*p* < 0.001) (Table 3). 

While all Colombian isolates showed no proteolytic activity (Pz ≥ 1), both Vancouver Island strains showed low proteolytic activity, with values of Pz of 0.82 and 0.78, for VGIIa and VGIIb, respectively. Similarly, all Colombian isolates showed medium phospholipase activity, with an average Pz = 0.62, which was similar to the activity presented by the VGIIb strain, which showed a Pz = 0.68, and lower than the activity presented by the VGIIa strain, which showed a Pz = 0.7 (Table 3). 

As expected, all isolates from Colombia and Vancouver Island showed urease activity, however, with no statistically significant difference among the studied isolates (*p* = 0.87).

### 3.5. Isolates from Cucuta Are of Considerable Virulence

Based on the phenotypic results obtained, the following strains from Colombia were selected to carry out in vivo virulence studies: H0058-I-357, H0058-I-1511, H0058-I-2792, and H0058-I-3030. Survival curves showed that the VGIIa isolate ENV152 from Canada was the most virulent, with median survival time of six days, as reported previously [10]. Following this, H0058-I-1511, H0058-I-2792, RB28 (VGIIb), and H0058-I-357 presented median survival times of 16, 16, 19, and 31 days, respectively. By the end of the experiment, H0058-I-3030 killed only one mouse, so it was considered the least virulent isolate (Figure 4). The results obtained by histopathology and recovery of CFU from brain, lung, and spleen, showed a greater migration of blastoconidia to the brain. However, the greatest damage was observed in the lungs, causing the death in mice mainly due to the cause of pulmonary cryptococcosis and organ failure. Another important finding was that there was no trend or any significant correlation between the virulence of the strains and the ability to migrate to the brain, lung, and spleen, since there was no difference in the number of CFU/g (*p* = 0.89) recovered from these organs in the six isolates tested (Appendix A).

## 4. Discussion

The present study identified and characterized a highly clonal population of *C. gattii* VGII isolates that were recovered from cryptococcal meningitis cases in Cucuta, a region with a significant incidence of cryptococcal infection in Colombia, during a period of fifteen years. Of great concern is the fact that most of the studied isolates caused disease in otherwise healthy hosts, as 11 out of the 13 patients did not have any apparent condition that may have increased the risk of cryptococcosis, which in turn supports various studies that have regarded *C. gattii* as a primary pathogen [1]. Interestingly too, this study included an isolate recovered from an eight-year-old girl, which is a very uncommon finding considering that not only in Colombia but worldwide the presentation of this mycosis is very rare in children [45], except in North Brazil [46], yet it shows the broad range of hosts that *C. gattii* can affect. Importantly, the MLST genotype (ST25), in which most of the studied isolates clustered (84.6%), has not only been reported in six other departments in Colombia [22], but also in Aruba [32] and Venezuela [23], and not only affects humans but also other animals, as the isolate from Aruba (WM 06.33) is a veterinary case recovered from a goat in 1953. In addition, ST25 has also been identified in our lab from environmental samples in Bogota, Colombia, which together suggests an intercontinental and old circulation of this genotype and its association with environmental reservoirs.

The present study also showed that although the isolates from Cucuta have different MLST profiles when compared to the genotypes from the VGIIa and VGIIb isolates previously described from the Vancouver Island outbreak, which occurred in the late 1990s [10,12] (Figure 1, Table 2), carry some of the same MLST alleles. As MLST analysis is currently the globally standardized typing method for *C. neoformans* and *C. gattii* [5], the results from this study contribute new data to the global population genetic analysis of these yeasts. The identification of highly clonal isolates by MLST is significant, as clonal dispersion has shown to be an important characteristic of outbreak-related cases, not only on Vancouver Island and in the PNW of the USA, but also in Australia and Thailand [13,23,32,34,47].

It is of notice, that after whole genome sequence analyses reported in a previous study [23], the gene regions specific to the highly virulent VGIIa genotype from Vancouver Island and the PNW were only found in three other global subtypes, originating from Brazil, Colombia, and Australia, with the isolate from Colombia, specifically the one from Cucuta, belonging to the ST25 (H0058-I-1278 = WM 05.275). These VGIIa-specific regions were mostly in a contiguous gene cluster and were rarely seen in other global subtypes, suggesting a greater likelihood of novel genes [23]. Although the isolates causing the outbreak in North America have distinct MLST subtypes from those from Cucuta, and are separated by WGS from this and other Colombian isolates, there is evidence of recombination among them and among other isolates from South America, for instance, the ST25 isolate from Cucuta shares many of their linked loci with an isolate from Brazil (WM 05.529 = LMM855) (ST27) [23].

In addition to the genotypic investigations, the isolates from Cucuta were also studied and compared phenotypically with the Vancouver Island VGIIa and VGIIb strains to determine similarities or differences among them. When comparing the growth under different temperature regimes, it was found that despite the dissimilarity in the average climatic conditions of both regions (Cucuta 30–35 °C and Vancouver Island 3–20 °C), the isolates had similar growth rates, showing that *C. gattii* has a wide adaptability to changes in temperature conditions, which can allow the fungus to have a certain degree of adaptation to the environment and subsequent colonization of the host. When the isolates were grown under different relative humidity conditions, it was evident that the isolates were capable of growing under minimal moisture conditions (20% relative humidity) without morphological abnormalities in the colony morphology. Other studies in some fungi, such as *Candida albicans*, have shown that they are susceptible to desiccation, and that under conditions of drought they can quickly lose viability [37]. However, the *Cryptococcus* polysaccharide capsule confers the fungus the ability to significantly delay the drying process, allowing its survival at low levels of humidity. The amount of intracellular fluid retained, allows *Cryptococcus* in nature to adapt to fluctuations of relative humidity, given by temperature changes in the environment [48,49]. The increased growth rates observed in isolates of *C. gattii* included in this study, when the relative humidity was increased in the media, is consistent with results reported in environmental studies, which suggested that in nature, the time of year characterized by high rainfall and humidity, few hours of sunshine and low temperatures favor the incidence of *C. gattii*, mainly serotype B [50].

Although it is known that capsules provide yeast with protection against a dry environment and inhibits its phagocytosis in the host [51,52], large capsules and cells presented by the isolates from Cucuta were not associated with the virulence of the strains in the animal model when compared with the strains from Vancouver Island, which is in agreement with other reports that suggests that a thin polysaccharide capsule permits better crossing of the blood–brain barrier [53]. Another particular characteristic of the isolates from Cucuta is their mating type, as 12 of the isolates were mating type a and only one was MATα, while, not only the representative VGIIa and VGIIb strains, but all the Vancouver Island isolates have been reported to be mating type α [10]. A similar proportion of mating types have been already reported in Colombia, as well with *C. gattii* serotype B isolates [21], which gives more evidence of a low incidence of mating type α in the population of *C. gattii* serotype B in the country [21]. The mating type in *C. neoformans* and *C. gattii* has been reported in other studies as an aspect closely related to virulence, with isolates of the mating type α being more virulent, even though the association is not clearly understood [10,30]. As suggested with the Vancouver Island outbreak strains, the presence of only mating type a in the clonal ST25 population identified in Colombia may suggest the idea that same-sex mating could be the driving force for clonal reproduction and dispersion [32].

Isolates from Cucuta showed a greater activity of phospholipases with respect to the Vancouver Island isolates. However, a direct relation of the phospholipase activity with virulence was not observed for the Colombian isolates tested in the animal model, which is similar to the results reported by Huérfano et al., who evaluated the activity of phospholipase in environmental isolates of *C. neoformans* var. *grubii* and *C. gattii*, and did not find a direct relationship between the production of phospholipase and virulence [54]. Concerning the activity of the enzyme urease, although it has been reported as an important virulence factor for colonization of the host [43,54], in this study, a difference in the activity among the isolates evaluated was not observed. Nevertheless, with the mice model of infection it was possible to recognize that isolates of the genotype ST25 present a considerable virulence, especially when compared with the VGIIa strain, which is considered a highly virulent genotype (Figure 4).

Although South America has been proposed to be the major source of VGII diversity, with numerous subtypes and extensive recombination [23], it also appears that clones are emerging from it, including ST25, with phenotypic and genotypic characteristics that are similar to those of the isolates responsible for outbreaks of infection in other parts of the world. Our findings reemphasize the need for an ongoing surveillance of Colombian cryptococcal strains, especially in cryptococcosis endemic areas, to allow for an early detection of potentially highly virulent strains that can be spreading clonally, and to initiate a timely public health response to prevent the potential occurrence of similar outbreaks in Colombia and elsewhere.

## Figures and Tables

**Figure 1 jof-05-00032-f001:**
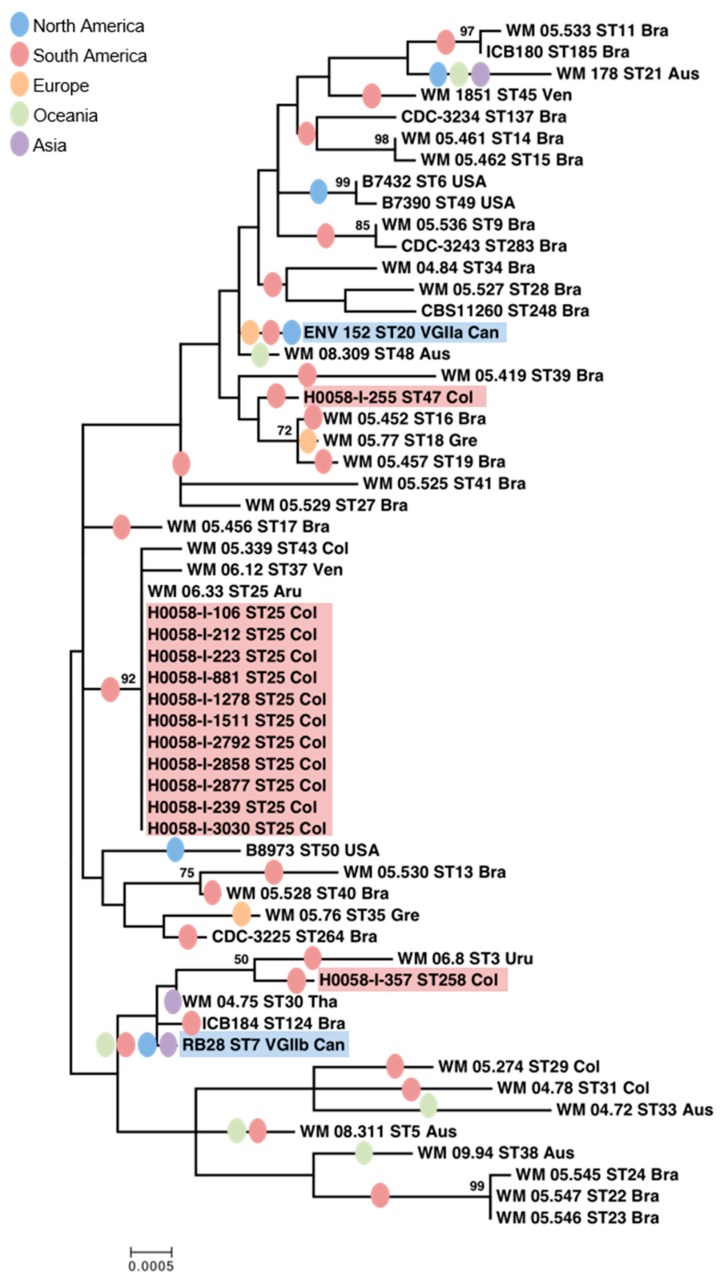
Dendrogram showing the genetic relationships between *Cryptococcus gattii* molecular type VGII isolates from Cucuta, the Vancouver Island outbreak (highlighted), and global isolates selected from previously published MLST studies [9,22,23,32,33,34]. The tree is based on maximum likelihood analysis of the concatenated seven ISHAM consensus MLST loci using the program MEGA 7.0.2631. Numbers on the branches indicate bootstrap values above 50. Isolates recovered in Aus: Australia, Bra: Brazil, Can: Canada, Col: Colombia, Gre: Greece, Tha: Thailand, Uru: Uruguay, USA: United States of America, and Ven: Venezuela. Ovals represent the continents where the sequence types have been previously identified.

**Figure 2 jof-05-00032-f002:**
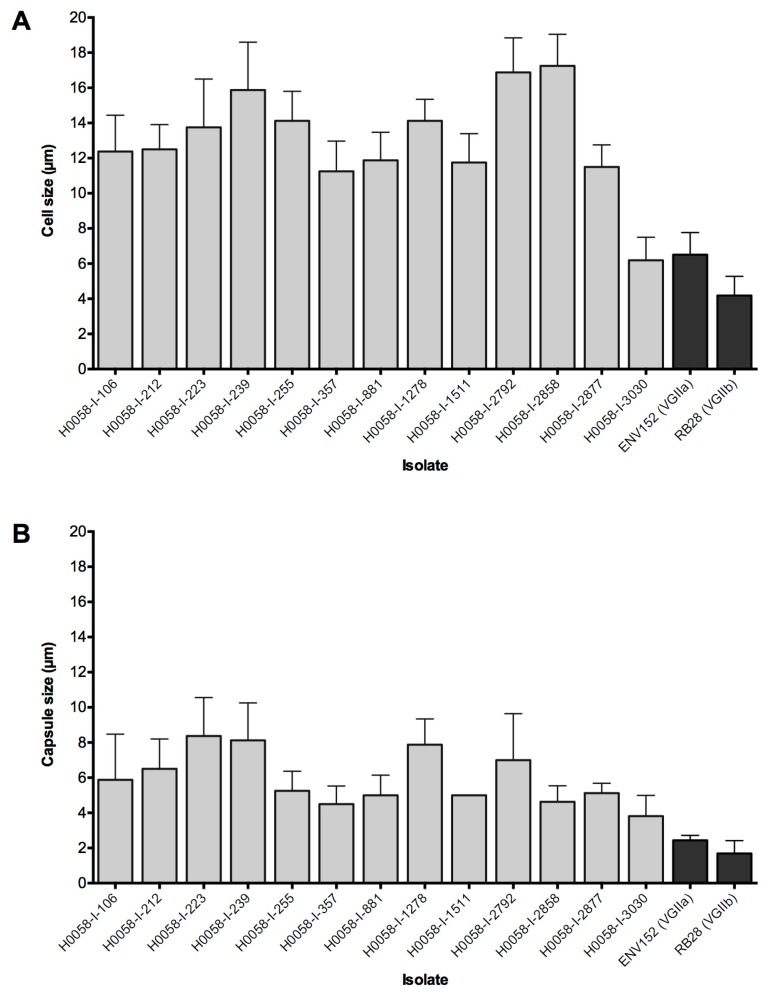
Distribution of the median and standard deviation of (**A**) yeast cell size and (**B**) capsule size of the studied *Cryptococcus gattii* VGII isolates. Light-grey and dark-gray columns represent isolates from Cucuta and Vancouver Island, respectively.

**Figure 3 jof-05-00032-f003:**
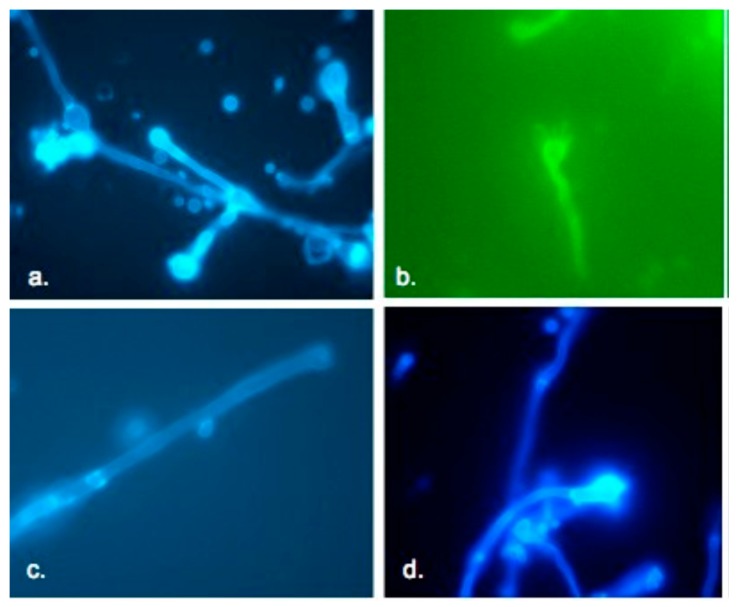
Sexual structures of *Cryptococcus gattii* VGII isolates that mated with an opposite mating type strain. (**a**) Presence of basidia resulting from the mating of *C. gattii* VGIIa (MATα) strain with JEC20 (*Cryptococcus neoformans* serotype D, MATa); (**b**) formation and detachment of basidiospores from basidium resulting from the mating of *C. gattii* strain H0058-I-357 (MATa) with JEC21 (*C. neoformans* serotype D, MATα); (**c**) hyphae with the presence of fibule junction; and (**d**) formation of basidiospores from a basidium resulting from the mating of *C. gattii* VGIIb (MATα) strain with JEC20 (MATa). All preparations were done with calcofluor (1 mg/mL) and observed at a wavelength of 425 nm, 40×.

**Figure 4 jof-05-00032-f004:**
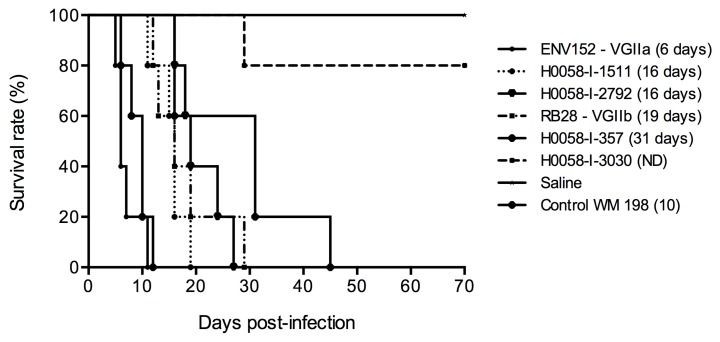
In vivo survival curves. BALB/c mice were inoculated with *Cryptococcus gattii* VGII isolates from Cucuta and Vancouver Island. Median survival time in days is indicated in brackets.

**Table 1 jof-05-00032-t001:** General information of clinical *Cryptococcus gattii* VGII isolates from Cucuta, Colombia.

Strain Number	Other Number	Year of Isolation	Patient’s Age (Years)	Patient’s Gender ^1^	Clinical Presentation	Outcome	Ref.
H0058-I-106	WM 08.290	1999	43	M	Meningitis	Living	-
H0058-I-212	WM 08.288	1993	54	M	Meningitis	Deceased	-
H0058-I-223	WM 08.289	1993	41	M	Meningitis	Living	-
H0058-I-239	WM 08.291	1993	8	F	Meningitis	Deceased	-
H0058-I-255	WM 08.292	1999	25	M ^2^	Meningitis	Living	-
H0058-I-357	WM 08.293	1995	11	M	Meningitis	ND	-
H0058-I-881	WM 08.295	1999	34	M	Meningitis	Living	[22]
H0058-I-1278	WM 05.275	2001	39	M	Meningitis	Living	[22,23]
H0058-I-1511	WM 05.399	2002	56	M	Meningitis	Living	[22]
H0058-I-2792	WM 08.297	2007	51	M	Meningitis	ND	[22]
H0058-I-2858	WM 08.298	2007	60	M	Meningitis	ND	[22]
H0058-I-2877	WM 08.299	2008	46	F ^3^	Meningitis	Deceased	[22]
H0058-I-3030	WM 08.305	2008	31	M	Meningitis	Living	-

^1^ M: male; F: female; ^2^ Transplanted patient using corticosteroids; ^3^ Patient with auto-immune disease using corticosteroids; ND: No data available.

**Table 2 jof-05-00032-t002:** Mating types, allele types (ATs), and sequence types (STs) of the studied *Cryptococcus gattii* VGII isolates. ATs and STs were identified through the MLST database at mlst.mycologylab.org.

Strain Number	Mating Type	*CAP59*	*GPD1*	IGS1	*LAC1*	*PLB1*	*SOD1*	*URA5*	ST
H0058-I-106	a	2	6	25	4	18	12	10	25
H0058-I-212	a	2	6	25	4	18	12	10	25
H0058-I-223	a	2	6	25	4	18	12	10	25
H0058-I-239	a	2	6	25	4	18	12	10	25
H0058-I-881	a	2	6	25	4	18	12	10	25
H0058-I-1278	a	2	6	25	4	18	12	10	25
H0058-I-1511	a	2	6	25	4	18	12	10	25
H0058-I-2792	a	2	6	25	4	18	12	10	25
H0058-I-2858	a	2	6	25	4	18	12	10	25
H0058-I-2877	a	2	6	25	4	18	12	10	25
H0058-I-3030	a	2	6	25	4	18	12	10	25
H0058-I-255	alpha	2	6	15	4	1	42	7	47
H0058-I-357	a	7	2	32	7	25	15	2	258
RB28 (VGIIb)	alpha	2	6	10	4	2	15	2	7
ENV152 (VGIIa)	alpha	1	1	4	4	1	14	7	20

**Table 3 jof-05-00032-t003:** Phenotypic characteristics of the studied *Cryptococcus gattii* VGII isolates growth at 37 °C.

Strain Number	Switching	Colony Morphology		Phenol-Oxydase (μg/mL)	Proteases		Phospholipases	
		Diameter (mm)	Texture		Mean (Pz)	Activity	Mean (Pz)	Activity
**Isolates from Cucuta**								
H0058-I-106	+	7.5	Smooth	196.3	1	None	0.60	Medium
H0058-I-212	+	9.5	Mucoid	35.3	1	None	0.57	Medium
H0058-I-223	-	12.0	Mucoid	216.2	1	None	0.67	Medium
H0058-I-239	+	7.6	Mucoid	153.9	1	None	0.63	Medium
H0058-I-255	+	7.0	Mucoid	1.2	1	None	0.60	Medium
H0058-I-357	+	10.0	Mucoid	193.1	1	None	0.68	Medium
H0058-I-881	+	10.3	Mucoid	185.4	1	None	0.63	Medium
H0058-I-1278	+	9.1	Mucoid	201.6	1	None	0.68	Medium
H0058-I-1511	-	7.1	Smooth	168.7	1	None	0.66	Medium
H0058-I-2792	+	8.4	Smooth	203.9	1	None	0.60	Medium
H0058-I-2858	+	8.0	Smooth	239.8	1	None	0.61	Medium
H0058-I-2877	+	9.3	Smooth	119.1	1	None	0.57	Medium
H0058-I-3030	+	7.8	Smooth	159.5	1	None	0.57	Medium
**Vancouver Island Isolates**								
ENV152 (VGIIa)	+	8.6	Smooth	150.82	0.82	Low	0.70	Low
RB28 (VGIIb)	+	8.3	Smooth	116.68	0.78	Low	0.68	Medium

NA: Not applicable.

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
