# Peer review of "Clonal Dispersal of *Cryptococcus gattii* VGII in an Endemic Region of Cryptococcosis in Colombia"

_jof, 2019, doi:10.3390/jof5020032_

Round 1
Reviewer 1 Report
In their study "Clonal dispersal of Cryptococcus gattii VGII in an endemic region of cryptococcosis in Colombia" authors describe the characterisation of VGII isolates from Columbia in comparison to isolates from the Vancouver Island Outbreak. There major finding is the relative abundance of a mating type and the potential consequences for this. This is an interesting study and their methodology relevant and thorough. I have some minor criticisms of interpretation of results, significance testing and English language. Please see specific comments below.
Specific comments:
1. L23-27 That last two sentences of the abstract are awkward and not terribly informative. It is stated much better in the discussion:
Our findings reemphasize
421 the need for an ongoing surveillance of Colombian cryptococcal strains, especially in cryptococcosis
422 endemic areas, to allow for an early detection of potentially highly virulent strains that can be
423 spreading clonally, and to initiate a timely public health response to prevent the potential occurrence
424 of similar outbreaks in Colombia and elsewhere.
2. L42 May I suggest adding the words "in our opinion" are added before" this taxonomic proposal"
3. L46-49 Is this due to geographical distribution or rather relative abundance (as the authors state later; see 5. below)?
4. L52 Change to:
"humans, domestic and wild animals. In 2003,"
5. L56 "Throughout" is an exaggeration change to 'several' or state how many or list the locations.
6. L67 Should be compared to similar surveiliance in other parts of the world.
7. L74 Define departments or use a more general term such as areas.
8. L76 Delete "Notoriously"
9. L159 Why use JEC21 and not use C. gattii mating pairs, also published by the Heitman lab?
10. L226 Be explicit about how they are closely related. It isn't number of isolates in each group, some simple mental arithmetic tells me a Fisher's test will not be significant.
11. Figure 2 a and b. Statistical significance testing required. One way ANOVA comparing the means of each column against the control VGIIa and VGIIb isolates.
12. L291 Add description of what Figure 3 shows. Do directly refer to figure in text.
13. L391 Primary sources should be cited here.
Author Response
Comments and Suggestions for Authors
In their study "Clonal dispersal of Cryptococcus gattii VGII in an endemic region of cryptococcosis in Colombia" authors describe the characterisation of VGII isolates from Columbia in comparison to isolates from the Vancouver Island Outbreak. There major finding is the relative abundance of a mating type and the potential consequences for this. This is an interesting study and their methodology relevant and thorough. I have some minor criticisms of interpretation of results, significance testing and English language. Please see specific comments below.
Specific comments:
1. L23-27 That last two sentences of the abstract are awkward and not terribly informative. It is stated much better in the discussion: “Our findings reemphasize the need for an ongoing surveillance of Colombian cryptococcal strains, especially in cryptococcosis endemic areas, to allow for an early detection of potentially highly virulent strains that can be spreading clonally, and to initiate a timely public health response to prevent the potential occurrence of similar outbreaks in Colombia and elsewhere”
The conclusion in the abstract was modified according to the reviewer’s suggestion.
2. L42 May I suggest adding the words "in our opinion" are added before" this taxonomic proposal"
We have clarified that the opinion on the “taxonomic proposal”, more than being our opinion, is the opinion of several researches. In fact, this is supported with the reference 7.
3. L46-49 Is this due to geographical distribution or rather relative abundance (as the authors state later; see 5. below)?
It is due to geographical distribution.
4. L52 Change to:
"humans, domestic and wild animals. In 2003,"
We have changed the sentence as suggested.
5. L56 "Throughout" is an exaggeration change to 'several' or state how many or list the locations.
We have changed the sentence as suggested.
6. L67 Should be compared to similar surveiliance in other parts of the world.
The prevalence of C. neoformansvar. grubbiin Colombia is comparable to that occurring worldwide. This was included in the text.
7. L74 Define departments or use a more general term such as areas.
The definition of department was included.
8. L76 Delete "Notoriously"
We have changed the sentence as suggested.
9. L159 Why use JEC21 and not use C. gattii mating pairs, also published by the Heitman lab?
This paper is part of a major grant in which a greater number of Vancouver strains were studied, included strain CDCR265 characterized as mating type alpha; this isolate was not used as a control strain for mating purposes since it was being also used as a tested strain.
10. L226 Be explicit about how they are closely related. It isn't number of isolates in each group, some simple mental arithmetic tells me a Fisher's test will not be significant.
We have explained why the Colombian isolates are more closely related to VGIIb.
11. Figure 2 a and b. Statistical significance testing required. One way ANOVA comparing the means of each column against the control VGIIa and VGIIb isolates.
One way ANOVA was done to compare the means of each column and the controls VGIIa and VGIIb
12. L291 Add description of what Figure 3 shows. Do directly refer to figure in text.
We have added a description in the text of what figure 3 shows.
13. L391 Primary sources should be cited here.
Primary sources were cited.
Reviewer 2 Report
Firacative and cols. characterized Cryptococcus gattii VGII isolates from Cucuta, an endemic region of cryptococcosis in Colombia, and compared these with representative isolates from the Vancouver Island Outbreak (VGIIa and VGIIb) using MLST analysis. Also, they evaluated the phenotypes (growth capacity under different conditions, morphology analysis, switching, mating type and enzimatic activity). Finally, virulence was studied in mouse model. This work is very relevant to cryptococcosis epidemiology, mainly due to potential risk of outbreaks.
Specific comment
Page 3 - Line 106 - Why in table 1 the authors listed the C. gattii isolates randomly? I suggest a list, ascending or descending, by year of isolation or strain ID. Also I strongly suggest to mantain the same order through the manuscript text.
Author Response
Comments and Suggestions for Authors
Firacative and cols. characterized Cryptococcus gattii VGII isolates from Cucuta, an endemic region of cryptococcosis in Colombia, and compared these with representative isolates from the Vancouver Island Outbreak (VGIIa and VGIIb) using MLST analysis. Also, they evaluated the phenotypes (growth capacity under different conditions, morphology analysis, switching, mating type and enzimatic activity). Finally, virulence was studied in mouse model. This work is very relevant to cryptococcosis epidemiology, mainly due to potential risk of outbreaks.
Specific comment
Page 3 - Line 106 - Why in table 1 the authors listed the C. gattii isolates randomly? I suggest a list, ascending or descending, by year of isolation or strain ID. Also I strongly suggest to mantain the same order through the manuscript text.
The isolates were listed ascendingly by strain ID as in Table 3.